# In-situ aerosol measurements at the Arctic Sammaltunturi measurement station during the Pallas Cloud Experiment 2022

John Backman<sup>1</sup>, Krista Luoma<sup>1</sup>, Henri Servomaa<sup>1</sup>, Ville Vakkari<sup>1</sup>, and David Brus<sup>1</sup> <sup>1</sup>Atmospheric Composition Research, Finnish Meteorological Institute, Helsinki, Finland **Correspondence:** john.backman@fmi.fi

#### Abstract.

This work describes the in-situ aerosol measurements at the Arctic Sammaltunturi measurement station in Pallas in northern Finland. This data paper, describes the instruments and the data post processing of key aerosol particle measurements that are relevant for cloud properties. Data reported here are part of the Pallas Cloud Experiment in 2022 (PaCE2022). The in-situ measurements described in this paper and are complementary to the research related to the PaCE 2022 campaign that investigates aerosol and cloud properties such as transects and profiles obtained from balloons, drones, and remote sensing techniques. All data from the campaign resides in a campaign dedicated data repository for easy access and overview. In addition, this data paper will also act as a future reference on how aerosol measurements are conducted and post-processed at the site for future

(last access: May 7, 2025). The data described here is available at https://doi.org/10.5281/zenodo.14900651 (last access: May 7, 2025) (Backman et al., 2025).

publications. The data set is available at PaCE 2022 campaign data repository at https://zenodo.org/communities/pace2022/

#### 1 Background

Aerosol particles and clouds are an intricate part of the climate system (IPCC, 2023). Aerosol particle suspended in air interact directly with solar radiation by scattering and/or absorbing solar radiation. How much aerosol particles interact with the sun's

- radiation directly depends on the optical properties of the aerosol particles and the albedo of the underlying surface (Haywood and Shine, 1995). Aerosol particles can also act as cloud condensation nuclei (CCN) or ice nuclei (IN) onto which water vapor can condense to form cloud droplets or ice crystals. The ability of an aerosol particle to form cloud droplets is influenced mainly by aerosol particle size, and to a lesser extent the aerosol particles chemical composition (e.g. Dusek et al. (2006) REF?). The number concentration of aerosol particles that can act as CCN is also of vital importance. An abundance of CCN active aerosol
- particles will form cloud droplets that are smaller in size which makes for brighter clouds, given that the amount of condensable water vapor stays the same (Twomey, 1974, 1977). Moreover, brighter clouds with smaller cloud droplets are less prone to form precipitation (Albrecht, 1989; Dagan et al., 2023). These aerosol-cloud interactions (ACI) are important for the Earth's climate as water vapor and clouds are way more important for the Earth's radiation budget than any non-condensable greenhouse gas.

The Arctic is a particularly vulnerable when it comes to climate change. Research has shown that the Arctic is warming four times faster than the rest of the world (Rantanen et al., 2022). Snow and sea ice are the most reflective surfaces that occur

naturally on Earth. These bright surfaces are able to reflect the sun's energy back into space, thus cooling the climate. Open water, on the other hand, is the most absorbing surface that occurs naturally on Earth. In the warming Arctic, snow and sea ice is replaced by open water, a highly reflective surface is replaced by a highly absorbing one, thus converting more of the sun's energy into heat. This phenomenon is known as Arctic amplification and is the reason that the Arctic is warming at such a fast rate (Serreze et al., 2009). The implications of these rapid changes currently happening in Arctic on aerosol and low-level

clouds have remained elusive.

Current aerosol modeling efforts reported the inherent limitations in deriving vertical aerosol distributions solely from surface-based measurements. As highlighted by Morrison et al. (2022), the vertical stratification of aerosols within the atmospheric column, particularly their relative positioning with respect to cloud layers, can have a significant impact on the sign

- of the radiative forcing, potentially inverting its sign. Also, large-scale climate models struggle simulating low-level Arctic mixed-phase clouds. A significant contributing factor to these discrepancies lies in the incomplete representation of aerosol microphysical properties, specifically those that could act as cloud condensation nuclei (CCN) and ice nucleating particles (INP). Under specific thermodynamic regimes prevalent in the Arctic, those factors were found to be limiting for cloud formation processes (Schmale et al., 2021).
- In the last two decades, extensive work has been done on ACI during Pallas Cloud Experiments (PaCE). Our previous research has shown that even the relatively clean air in Finnish subarctic is a complex mixture of potential aerosol precursors from various marine, biological and anthropogenic sources and that changes in anthropogenic emissions can have an impact on subarctic aerosol up to CCN sizes and also cloud micro-physical properties (e.g. Komppula et al., 2005; Kivekäs et al., 2009; Lihavainen et al., 2010; Asmi et al., 2011; Anttila et al., 2012; Filioglou et al., 2017; Gérard et al., 2019; Doulgeris
- et al., 2020, 2023). This paper is a data description paper for the in-situ aerosol measurements conducted during the Pallas Cloud Experiment (PaCE) in 2022 at Pallas measurement site in Northern Finland. This data is complementary to other data produced during the measurement campaign that includes vertical profiles and transects of a wide range of in-situ aerosol, cloud and remote sensing measurements, carried out on wide range of platforms including tethered balloons, drones, and UAVs. The duration of the PaCE 2022 campaign was from 15 September 2022 to 15 December 2022. An overview of the PaCE 2022 50 campaign is provided by Brus et al. (2025).
  - 2 Instrumentation

#### 2.1 Measurement location and inlets

55

Here we present the key physical aerosol particle parameters measured continuously at the Pallas Sammaltunturi measurements station (elevation 565 m a.m.s.l.). The station is located on top of a fell and is located at 67.97361°N 24.11583°E in the municipality of Muonio in the Lappland region in northern Finland (Hatakka et al., 2003). Due to the altitude of the station, the station is occasionally inside clouds in the fall and early winter.

A summary table of the instruments, the inlets they are connected to, and the means by which the sample aerosol is dried, is summarized in Table 1 and the inlets are shown in Fig. 1.

65

Figure 1. Pictures of the total inlet (a) and the  $PM_{2.5}$  inlet (b).

The in-situ aerosol instruments at the Sammaltunturi station are measuring through two aerosol inlets: a total inlet with no 60 cut-off, and an inlet with a cut-off of 2.5 µm. The inlets are located 2 m above the station's roof and about 6 m above ground. Both inlets are heated to avoid snow and ice buildup when the station is inside clouds or when it is snowing.

The total aerosol inlet is a custom built inlet and based on the design that was first deployed to the Jungfraujoch station in Switzerland (Weingartner et al., 1999). The design of Jungfraujoch inlet was received through personal communication and replicated at the Finnish Meteorological Institute's workshop. The inlet is essentially a big hood. The sample air is drawn from within the hood and down into the station through the roof. In addition to the inlet, the sample tube is also actively temperature controlled so that the sample air warms up before reaching the inside of the measurement station.

The second inlet is a total suspended particle (TSP) inlet by URG Corp. (USA). Downstream of the TSP inlet is a  $PM_{2.5}$  cyclone (MesaLabs BGI  $PM_{2.5}$  Sharp Cut Cyclone) which is also heated so that the cyclone won't get clogged by snow or ice.

After the inlets, in both of the measurement lines, the sample air is dried with a Naphion dryer (Permapure Monotube Dryer 70 MD-700-24S) and dry compressed air. The Naphion dryers are 60 cm long with a tube diameter of 17 mm. The dryers are located inside the measurement hut. The diffusional losses of the 60 cm long Nafion driers have been characterized to have an equivalent length of 2.5 m of conductive tubing when accounting for diffusional losses (Dick et al., 1995). The Nafion tubes are installed vertically under the inlets to avoid losses due to gravitational settling. The moisture permiates through the Naphion membrane from the sample air is flushed away with dry compressed air which is flowing in the opposite direction of

75 the sample air. This configuration maximizes the efficiency of the drier. The dew point of the compressed air is about -40 °C. The temperature (T) and relative humidity (RH) of the sample air, in both inlets, are monitored with Vaisala HMP110 sensors.

| Instrument   | Inlet | Dryer   | $\Delta t$ | ECAC       |
|--------------|-------|---------|------------|------------|
| CPC (3772)   | PM2.5 | Naphion | 1 s        | no         |
| DMPS         | PM2.5 | Naphion | 6.5 min    | 2021       |
| CCN          | PM2.5 | Naphion | 1 s        | no         |
| MAAP         | PM2.5 | Naphion | 1 s        | 2017       |
| APS          | Total | Naphion | 1 min      | no         |
| AE-33        | Total | Naphion | 1 min      | 2019, 2023 |
| Nephelometer | Total | Naphion | 5 min      | 2017, 2023 |
|              |       |         |            |            |

**Table 1.** Summary of the instruments with information on their inlet, drying method, their native time resolution ( $\Delta t$ ), and last participation on ECAC inter-comparison workshops.

In addition to the aerosol instruments, a Vaisala visibility sensor (model FD12P) is used to check if the station is inside clouds, or not. The visibility sensor is also mounted on the roof of the measurement building. In this work, the station is considered to be inside a cloud if the visibility is below 1 km. The data has been flagged accordingly.

# 80 2.2 In-situ instruments

The Pallas Sammaltunturi measurement station is part of the pan-European network for Aerosols, Clouds, and Trace Gases Research Infrastructure (ACTRIS) network, which provides funding, technical support, calibration and quality assurance through inter-comparison workshops, through the European Center for Aerosol Calibration and Characterization (ECAC). Not all instruments at the station are part of the ACTRIS network, but those who are have participated in the ECAC workshops either before or after the PaCE 2022 campaign.

### 2.2.1 CPC

Total particle number concentrations (PNC) are measured with condensation particle counters (CPCs). At Pallas, the CPCs use buthanol as a working fluid. Butanol is used to increase the size of the particles for subsequent detection by optical means. The CPCs are laminar flow CPCs where all the sample aerosol flows through a warm saturation chamber, which saturates the

- sample air with butanol vapor. The sample is then cooled in a diffusion-type cooling section where the aerosol particles grow by condensation to optically detectable sizes. The growth through condensation is achieved by super-saturation of the butanol vapor in the cooling section. An overview of the development of these kinds of CPCs and their detailed working principles is given by McMurry (2000). The CPC to measure total particle number concentrations at Pallas after the PM2.5 inlet is a TSI Inc. model 3772 buthanol CPC (Hermann et al., 2007). The 3772 CPC's 50% cutoff diameter is around 7 nm with the standard
- operating parameters (Tuch et al., 2016), namely the temperature difference between the saturator and the condenser.

# 2.2.2 DMPS

Particle number size distributions (PNSD) are measured with custom built Differential Mobility Particle Analyzers (DMPS). The aerosol particles are selected using a Vienna-type Differential Mobility Analyzer (DMA, Winklmayr et al. (1991)) and PNC after the DMA is measured with a CPC (TSI Inc. model 3010). In the DMA, particles are selected by their electrical
mobility in an electrical field as they flow through the DMA, Knutson and Whitby (1975). Since the DMA measures electrical mobility, the particles need to have a charge. Hence, before the DMA, a Ni-63 radioactive source is used as a neutralizer and charger to achieve a known charge distribution of the aerosol (Wiedensohler, 1988). A known charge distribution is achieved using a bipolar charger of Ni-63 with an initial activity of 370 MBq. The probability of charged particles to pass through the DMA's electric field is described by the DMA's transfer function, e.g. Zhang and Flagan (1996). The transfer function, without considering diffusion, is a function of the flow rates through the DMA, Flagan (1999). The flow rates have been chosen so that the account for the short flow rates have been chosen so that the account for the short flow rates through the DMA, Flagan (1999).

the aerosol-to-sheath flow rate has a ratio of 1/5 which give a broad transfer function, but allows for the detection of particles up to 800 nm in size. The smallest size of the DMPS is 10 nm.

# 2.2.3 CCN

The Cloud Condensation Nuclei (CCN) were measured with a continuous-flow streamwise thermal-gradient CCN counter (CCNc-100, e.g. Rose et al. (2008)), manufactured by Droplet Measurement Technologies, Inc. (DMT). The CCNc-100 operates on the principle of creating a supersaturated environment to CCN into droplets. The core of the instrument is a continuousflow thermal-gradient diffusion chamber, where a controlled temperature gradient is established. Since water vapor diffuses faster than heat, the temperature gradient results in a supersaturated environment inside. Aerosol particles are drawn into the chamber where they activate at the preset level of supersaturation uptake water vapor and grow into droplets. At the bottom of the chamber is an optical particle counter (OPC) that is used to determine the size and number of the droplets by intensity of

light scattering (Droplet Measurement Technologies, Manual for Single-Column CCNs, DOC-0086 Revision M, 2017). The CCNc was connected to the total inlet to be able to sample also dried cloud activated aerosol during the station in-cloud periods. The CCNc setup utilized a three-way valve that allowed switching between total and size resolved sampling of aerosol by CCNc-DMA . In parallel to CCNc and after CCNc-DMA, the total particle number concentration was measured with CPC

120 (model 3010, TSI Inc.). Similarly as above, the CCNc-DMA setup used a bipolar charger of Ni-63.

# 2.2.4 APS

An aerodoynamic particle sizer (APS) measures the aerosol size distribution in the size range of 0.5 - 20.5 μm based on the aerodynamic diameters of the particles (B. T. Chen and Yeh, 1985). Therefore, the APS size range covers the coarse mode particles, which are typically classified as particles bigger than 1 μm in diameter. Due to their size, the coarse mode particles, although small by numbers, can contribute substantially to total particle mass (*PM*), and aerosol optical properties.

The basic operating principle of the APS is to measure the time-of-flight of individual particles in an accelerating air flow. The aerosol particle sample is accelerated in a nozzle and the time-of-flight of the particles in the flow are detected by two

laser beams. Due to their inertia, the aerosol particles do not accelerate as quickly as the air flow, and the time-of-flight of the particles differs between particles because of their mass, shape, density and size. The aerodynamic diameter is defined as the
diameter of a spherical particle with a density of 1000 kg m<sup>-3</sup> that has the same settling velocity as the particle in question. The time-of-flight data of the individual particles are converted to the aerodynamic diameter based on a calibration with polystyrene latex spheres that have known size and density.

For the APS to successfully measure the time-of-flight of an individual particle, the particle has to pass both of the lasers beams and cause two a large enough signals within a certain time window. Invalid measurements are caused by too large particles, particles detected by only one of the laser beams, and by particle coincidence in the measurement volume.

Instrument specific uncertainties and unit-to-unit variability are caused by inaccurate sample and sheath flow rate, the alignment of the flow and the lasers, and impurities in the optics cause uncertainties and unit-to-unit variability. Pfeifer et al. (2016) reported that the unit-to-unit variability of 15 APS instruments is about 10-20 % for particles sized 0.9-3  $\mu$ m. For smaller and larger particles the variation increases considerably and they recommended to use caution when applying data below 0.9  $\mu$ m or above 3  $\mu$ m. Another study by Vasilatou et al. (2022) reported high counting efficiencies between 0.7-5  $\mu$ m, but it included

140

135

only a few instruments in the comparison.

#### 2.2.5 Nephelometer

An integrating nephelometer measures the light-scattering coefficient ( $\sigma_{sp}$ ) that describes the amount of scattered light over a unit path length. The  $\sigma_{sp}$  is reported in units of Mm<sup>-1</sup>. At Pallas, the Nephelometer is a TSI Inc. model 3563 Nephelometer (Anderson et al., 1996) measuring at three wavelengths (450, 550, and 700 nm). A review on the development of integrating nephelometers to measure light scattering by aerosol particles is provided by Heintzenberg and Charlson (1996). In brief, the working principle of an integrating nephelometer is that sample air is drawn into a detection chamber where it is illuminated. The illumination is done through a diffusor glass who's surface is perpendicular to the surface of the detector. This geometry provides a cosine weighted illumination of the sensing volume which is desired to derive the  $\sigma_{sp}$  (Eq. 2.3. in Heintzenberg and Charlson (1996)). By shadowing the part of the measurement cell that represents forward scattering angles (

In practice, the integrating nephelometer measures light-scattering by both the air molecules and the aerosol particles. To obtain the amount of scattering by the aerosol particles alone, the amount of scattering by gases is subtracted from the measurements of sample air. This subtraction is possible by regularly measuring particle free air. To calibrate the instrument, the measurement chamber is filled with a calibrating gas with well-known scattering characteristics (e.g.,  $CO_2$ ). Because light

160 scattering depends on the wavelength ( $\lambda$ ), the measurements are optimally conducted at several wavelengths either by using discrete light sources or detectors.

# 2.2.6 MAAP

The Multi-Angle absorption photometer (MAAP, Thermo Sci. model 5012) is a filter-based light absorption photometer. Filter based absorption photometers measure the change in optical properties of the filter tape as aerosol particles deposit onto the filter. The MAAP has three detectors that are used to derive the light absorption coefficient ( $\sigma_{ap}$ ) of the aerosol. The  $\sigma_{ap}$  is calculated from the change in the filter properties during a time window of  $\Delta t$ . One detector is used to measure the transmittance of light through the filter, whereas the two other detectors are used to measure the back-scattered light from the filter at multiple (two) angles (Petzold and Schönlinner, 2004; Petzold et al., 2005). The filter transmittance (Tr) is defined as  $I/I_0$  where I is the intensity of light transmitted through the filter and  $I_0$  is the light intensity transmitted through a pristine filter. The back-scattering measurements of the filter is the used in a radiative transfer model to distinguish between diffusely

and Gaussian scattered light in the backward direction.

In the MAAP, the Tr and back-scatter measurements are used to account for nonlinearities in the instrument's response as the filter gets laden with aerosol particles. The nonlinearities are compensated for by solving a radiative transfer model using the measurements made on the filter (Petzold and Schönlinner, 2004). The MAAP is a one wavelength instrument and reports

equivalent black carbon (eBC, Petzold et al. (2013)) at a wavelength of 637 nm (Müller et al., 2011). The measured  $\sigma_{ap}$  values are converted into eBC mass concentrations using a fixed mass absorption cross section (MAC).

#### 2.2.7 AE-33

The Aethalometer (model AE-33, Magee Scientific) is also a filter-based absorption photometer. The AE-33 derives  $\sigma_{ap}$  from changes in Tr during the time  $\Delta t$  as aerosols are deposited onto the filter on two sample spot areas. The two spots are used to compensate for the nonlinearities in the instruments response as aerosol particles are deposited onto the filter. In the AE-33,

180

to compensate for the nonlinearities in the instruments response as aerosol particles are deposited onto the filter. In the AE-33, a light source illuminates one side of the filter while the detectors are on the other side of the filter measuring the Tr of both sample spots simultaneously. In the AE-33, the Tr measurements are reported as filter attenuation (ATN) values as

$$ATN = -100 \ln\left(\frac{I}{I_0}\right). \tag{1}$$

In the equation above,  $I/I_0$  is the same as Tr. The ATN values are calculated separately for both sample spots, spot 1 and spot 2. The factor of 100 applied for numerical convenience.

As the two spots on the fiber-filter get laden with light absorbing aerosol particles, ATN values increase; i.e. less light reaches the two detector on the opposite side of the light source. The AE-33's dual spot correction algorithm compensates for the non-linearity in the attenuation decrease as particles deposit onto the filter by comparing the change in ATN between the two spots. In brief, this is possible since the flow rate of the two spots are different which means that the accumulation of

aerosol particles onto the two different spots differ (Drinovec et al., 2015). The different rate of change in *ATN* for these two spots, for concurrent measurements, is used to calculate the light absorption coefficient without the non-linear filter loading effects. One additional spot, which does not get loaded with aerosol particles, is used to monitor the intensity of the light source and is not affected by the sample aerosol and is called the reference spot.

The AE-33 operates at 7 different wavelengths (370, 470, 520, 590, 660, 880 and 950 nm). Similar to the MAAP, σ<sub>ap</sub>
measurements are also reported as eBC mass concentrations by the AE-33 firmware using fixed MAC values. Regular 'zero-flow' and optical tests are automatically run to account for any changes in detector response or light source intensity. When the filter loading has reached a attenuation (*ATN*) value of 120, the Aethalometer invokes a filter change.

#### **3** Data processing

In the data set, data is reported in UTC 0 h as daily files starting at midnight. All data reported has been converted to standard temperature (*T*) and pressure (*p*), which is 273.15 K and 101.325 kPa, henceforth referred to as STP conditions, except the CCN data which is in ambient condition. Although the data is reported at STP conditions, the flow rates (*Q*) reported for the instruments in this section can be in either STP conditions in liters per minute (lpm) or at ambient conditions as volumetric liters per minute (vlpm). All data with a relative humidity (*RH*) above 40% has been removed, except for the CPCs measuring total PNC. The *RH* data as measured in the sample air in the total and PM<sub>2.5</sub> inlet is included in the data files so that the user can omit those data afterwards if desired. Furthermore, when the visibility at the station is below 1 km, a flag has been added

to the data to show that the station was inside a cloud.

The data is quality assured and corrected for known measurement artifacts while keeping the original time resolution. Invalid measurements (e.g., periods of maintenance or malfunctions) are omitted from the data. The flow rates of the instruments are measured periodically, during site maintenance and is recorded in the station's logbook (not shown).

210

The sample flow rate of the instruments can directly affect the measurements and the concentrations they report. This holds true for the MAAP, AE-33, CPC, DMPS's CPC, and APS. For these instruments, a flow correction factor  $F_{\rm flow}$  has been applied as follows

$$F_{\rm flow} = \frac{Q_{\rm inst}}{Q_{\rm meas}} \tag{2}$$

where  $Q_{\text{inst}}$  is the flow rate that the instrument thinks it is measuring with, and  $Q_{\text{meas}}$  is the measured flow rate that is 215 periodically measured with an external and high-accuracy bubble-flow meter at the inlet port of the instrument.  $F_{\text{flow}}$  is then the factor by which the concentrations will be multiplied with to get the true concentration.

The naming convention is described by Brus et al. (2025). In brief, all data described in this work is saved as .csv ASCII text format. The file names for each instrument is described separately for each instrument. The following convention is used: YYYY stands for the year, MM for the month, and DD for the day, all in numerals and with zero padding.

220

- Included in the data set is the automatic weather system (AWS) data from the station. The data is in the aws\_PaCE2022.zip file and the file names inside the zip archive are named FMI.AWS.a1.YYYYMMDD.csv. The files comprise 11 columns: datetime, Temp 570m (C), Dew point temp (C), Humidity 570m (%), Pressure (hPa), Wind speed (m/s), Wind dir (deg), Sun rad 1 (W/m2), Sun rad 2 (umol/s/m2), FD12P visibility (m), and Rain intensity (mm/h). The datetime columns shows the end time of the sampling period in as YYYY-MM-DD HH:MM:SS. Temp 570m (C) is the ambient temperature at the station in <sup>5</sup>C. Data point temp (C) is the ambient data point temperature in <sup>6</sup>C. Pressure (hPa) is the ambient pressure in hPa. Wind aread
- 225 °C, Dew point temp (C) is the ambient dew point temperature in °C, Pressure (hPa) is the ambient pressure in hPa, Wind speed