# Peer review of "In-situ aerosol measurements at the Arctic Sammaltunturi measurement station during the Pallas Cloud Experiment 2022"

_Earth System Science Data, 2025_

## Referee Comment (RC1)

Overall evaluation:

In this manuscript, the authors present a comprehensive description of the instruments and the measured dataset that are complementary to the research related to the PaCE 2022 campaign that investigates aerosol and cloud properties. The instrument and datasets are of good novelty and usefulness, and the manuscript is well written. There are no major comments from me, but a few technical corrections and suggestions. I would recommend a minor revision from the authors to make the corresponding corrections before the manuscript gets published.

Specific comments & technical corrections:

1. Line 18. The reference citation shown as (e.g. Dusek et al. (2006) REF?) should be corrected.

2. Line 23. The statement of 'water vapor and clouds are way more important for the Earth's radiation budget than any non-condensable greenhouse gas' is kind of oversimplified which would possibly lead to a misleading that greenhouse gases are not important for the radiation budget. The authors are recommended to provide a little more detail on how water vapor, as the primary greenhouse gas, dominates the Earth's radiation budget, while non-condensable greenhouse gases act as critical forcings but not as important as water vapor and clouds.

3. Line 79. Certain conditions like heavy snowfall, especially when accompanied by strong winds, could also reduce the visibility to below 1 km. Would this affect the measurement when the station is flagged as 'inside a cloud' when it's not?

4. Line 100. The citation Knutson and Whitby (1975) should be written as (Knutson and Whitby, 1975). Same for 'Flagan (1999)' in line 105.

5. Line 122. For' aerodoynamic', do you mean 'aerodynamic'?

6. Line 133-134. It should be written as 'the particle has to pass both of the laser beams and cause two large enough signals within a certain time window.'

7. Line 158. With 'the subtraction is possible by regularly measuring particle free air', is there a companion instrument that is able to clear out the particles from the sample air in order to measure the particle free air? Or is it only a theoretical assumption?

8. Line 174, should 'one wavelength instrument' be 'single wavelength instrument'?

9. Line 406, 'achieve the final data' should be 'obtain the final data' or 'acquire the final data'.

10. For Figure 6, it is recommended to modify the sizes of labels along the axis by enlarging the one along the x-axis while shrinking the one along y-axis. The label sizes in Figure 7 are more suitable.

11. I would personally recommend that the authors add a brief and general summary of the components of the instrument and data format to the 'Summary' section. The current subsection appears to be more of a directory while not providing much of actual 'key takeaways'.